# Three-Dimensional Organization of Chicken Genome Provides Insights into Genetic Adaptation to Extreme Environments

**DOI:** 10.3390/genes13122317

**Published:** 2022-12-09

**Authors:** Dan Shao, Yu Yang, Shourong Shi, Haibing Tong

**Affiliations:** 1Laboratory of Poultry Production, College of Animal Science, Shanxi Agricultural University, Jinzhong 030801, China; 2Poultry Institute, Chinese Academy of Agricultural Sciences, Yangzhou 225125, China

**Keywords:** three-dimensional genome, chicken, topologically associated domains, A/B compartments, extreme environments

## Abstract

The high-throughput chromosome conformation capture (Hi-C) technique is widely used to study the functional roles of the three-dimensional (3D) architecture of genomes. However, the knowledge of the 3D genome structure and its dynamics during extreme environmental adaptations remains poor. Here, we characterized 3D genome architectures using the Hi-C technique for chicken liver cells. Upon comparing Lindian chicken (LDC) liver cells with Wenchang chicken (WCC) liver cells, we discovered that environmental adaptation contributed to the switching of A/B compartments, the reorganization of topologically associated domains (TADs), and TAD boundaries in both liver cells. In addition, the analysis of the switching of A/B compartments revealed that the switched compartmental genes (SCGs) were strongly associated with extreme environment adaption-related pathways, including tight junction, notch signaling pathway, vascular smooth muscle contraction, and the RIG-I-like receptor signaling pathway. The findings of this study advanced our understanding of the evolutionary role of chicken 3D genome architecture and its significance in genome activity and transcriptional regulation.

## 1. Introduction

Chickens were domesticated from the red jungle fowl subspecies Gallus spadiceus approximately 6000 to 8000 years ago in South and Southeast Asia [1,2,3]. Subsequently, plenty of indigenous breeds were formed after extensive natural and artificial selection and have developed genetic adaptations to a wide range of eco-geographic conditions, particularly the breeds in frigid and tropical regions and high-altitude regions [4]. As is widely known, environmental pressure is a considerable driver for the shaping of the animal genome; therefore, genomic differential detections can help elucidate the genetic basis of acclimation to varying environmental conditions and afford profound understandings of functionally genetic variants [5]. Meanwhile, considering their short growth and reproductive periods as well as wide distribution, chickens are ideal models for studying genetic adaptations to various environments [6].

According to information, genetic adaptations to extreme environments in chickens have attracted many researchers, but most studies have focused primarily on high-altitude adaptations in Chinese chickens [7,8] and the adaptations to arid and hot environments in African and Asian chickens [9,10] using chip or whole-genome analyses. Meanwhile, few studies have focused on the Chinese chicken adaptation to frigid and tropical environments, despite the presence of diverse local Chinese chicken breeds. In consideration of current and future climate impacts on the global environment [9], the genetic footprint of adaptations to tropical and frigid climates in Chinese chickens is critical for the modern chicken industry.

In this study, to better understand the genetic footprints of extreme environmental adaptation, we used an integrated analysis combining liver high-throughput chromosome conformation capture (Hi-C) data [11] with the liver transcriptome sequencing of Lindian chicken (LDC) and Wenchang chicken (WCC) from two extreme environments (more frigid to more tropical environments). We generated genome-wide profiles of chromatin interactions using Hi-C technology, which was performed to characterize spatial chromatin compartments and evaluate the DNA interaction frequencies at a resolution from several dozen kilobases to megabases [12,13]. We identified the characteristics of A/B compartments as well as topologically associating domains (TADs) in two different sources of liver cells. We also observed switched compartmental genes (SCGs) and enriched pathways associated with the variant 3D genome, which indicated the genetic adaptations to tropical and frigid environments.

## 2. Materials and Methods

### 2.1. Sample Collection and Sequencing Using Hi-C

Based on the findings from previous studies [14,15], liver samples were collected from one Wenchang chicken (a laying hen of 43 weeks from Wenchang, Hainan, 32 °C) and one Lindian chicken (a laying hen of 42 weeks from Lindian, Heilongjiang, −3 °C) from national natural poultry conservation farms with tropical and frigid environmental zones, respectively. All of these birds were reared in a semi-open chicken house and fed with a traditional corn–soybean meal diet. The animal tests we conducted were in accordance with the guidelines provided by the animal care and use committee of the Poultry Institute, Chinese Academy of Agricultural Science (Approval No. 2020-1103, Yangzhou, Jiangsu, China). The chicken livers were fixed in 1% formaldehyde solution with MS buffer (0.1 M sucrose, 50 mM NaCl, 10 mM potassium phosphate, pH 7.0) under vacuum for 30 min at room temperature. Next, the livers were cultivated under vacuum at room temperature for 5 min in MC buffer supplemented with 0.15 M glycine. Following this, we used liquid nitrogen to homogenize approximately 2 g of fixed tissue and resuspended it with nuclei isolation buffer. Next, we filtered the solution with a 40 nm cell strainer to obtain the liver cells. Following this, the procedure for concentrating the nuclei from the flowthrough and the subsequent denaturation process was completed on the basis of a 3C protocol built for maize.

The chromatin was digested using 400 U HindIII restriction endonuclease at 37 °C for 16 h. Following this, the DNA ends were labeled with biotin and incubated for 45 min at 37 °C. Next, DNA ligation was conducted by adding T4 DNA ligase and incubating for 4 to 6 h at 16 °C. After ligation, proteinase K was added to bottom up the crosslinking process during overnight incubation at 65 °C. Then, we skimmed the DNA fragments and dissolved them using 86 μL water. We removed the unligated ends, fragmented the skimmed DNA to 300–500 bp, and then repaired the DNA ends. DNA fragments labeled using biotin were eventually captured using Dynabeads^®^ M-280 Streptavidin (Thermo Fisher Scientific, Waltham, MA, USA). We subjected Hi-C libraries to quality control and sequenced them with an Illumina HiSeq X Ten sequencer.

### 2.2. Mapping and Filtering of Hi-C Read and Contact Matrices Generation

First, low-quality paired reads (reads with ≥10% unidentified nucleotides (N); >10 nt aligned to the adaptor, which allowed for ≤10% mismatches, >50% bases with phred quality < 5, and putative PCR duplicates generated in the library construction process) were removed; the low quality was primarily due to adaptor contamination and base-calling duplicates. Following this, we mapped the high-quality paired-end Hi-C reads to Gallus_gallus-5.0 and filtered them using HiCUP v0.5.10 [16]. HiCUP eliminated the sequences representing the trial Hi-C artifacts and other uninformative di-tags, as the presence of even a few unusable ditags can yieldimproper outcomes for the genomic structure. We divided the genome into 1Mb bins, and computed the read pair numbers within two regions based on those scanning interactions using hicPipe [17]. Next, those expected interactions were calculated using this software. We counted the norm interactions using the scanning interactions divided by the anticipated interactions. The criterion interactions were used for every two bins to generate the criterion contact matrix.

### 2.3. Identification of Compartment A/B

The identification of A and B compartments is a standard procedure for the analysis of Hi-C data and represents the application of a principal component analysis of the interaction matrix [18]. A and B compartments have been previously shown to correspond relatively well to active and inactive genomic regions [19]. The positive or negative values of the first principal component separate chromatin regions into A/B spatially segregated compartments. The compartment A/B was identified using the 1 Mb interaction matrix at a 500 kb resolution to explore the chromatin compartment types with R-package HiTC as formerly described [20]. The eigenvector values of the two Hi-C samples were calculated using hiclib [21]. Bins with positive values were defined as compartment A, and those with others were defined as compartment B. The distribution of the E1 value of genes in each bin was calculated.

### 2.4. Generation of Interchromosomal Contact Matrix

The anticipated number of interchromosomal interactions for each chromosome pair i, j was calculated by multiplying the fraction of interchromosomal reads including i with the fraction of interchromosomal reads containing j and multiplying the value with the t total amount of interchromosomal reads. Next, we calculated the abundance using the real number of interactions scanned between i and j and divided the value by the anticipated value. We calculated that interchromosomal connection probability using the scanning reads pair numbers between the chromosome pair i, j, parting it via the anticipated value. Then we calculated the anticipated number of interchromosomal interactions for each chromosome pair i, j by multiplying the proportion of interchromosomal reads embracing i with the ratio of interchromosomal reads containing j and the total number of interchromosomal reads.

### 2.5. Identification of TADs and TAD Boundaries

The normalized contact matrix was used as input to appraise TAD as recorded in a mammalian model [22]. Next, the directionality index (DI) was computed from 2 Mb upstream to 2 Mb downstream along the center of each bin at a 40 kb resolution, and then we used the Hidden Markov model (MATLAB 2013, HMM_calls.m) to forecast the states of DI for final TAD generation. Next, we used the same criteria (a distance of 400 kb between two adjacent TADs) to differentiate between unorganized chromatin with topological boundaries, i.e., the topological boundaries would be less than 400 kb and unorganized chromatin would be greater than 400 kb. The transcription start sites (TSS) were computed based on the number of annotated genes. First, the position of the TAD boundary center was identified, following which the number of TSS in each 10 kb window upstream and downstream of the TAD Boundary Center with the distance delimiting size of 10 kb was calculated.

### 2.6. Boundary Correlation Experiments

The boundary correlation was implemented as formerly reported [22]. We incorporated the mid position of the boundaries between two experiments of interest and computed the directionality indexes ± 10 bins for each center. Following this, 20 bins from each of the two cell lines were randomly selected and the Spearman associations between the two vectors were calculated. After that, we obtained the Spearman association coefficient distribution after repeating the randomization 10,000 times.

### 2.7. Transcriptome Sequencing and Analysis

Liver tissues from eight chickens (four LDCs of 42 weeks from Lindian, Heilongjiang and four WCCs of 43 weeks from Wenchang, Hainan) were collected from the national natural poultry genetic resource conservation farm and subjected to sequencing for transcriptomic analysis. We developed sequencing libraries with NEBNext^®^ UltraTM RNA Library Prep Kit (NEB, Ipswich, MA, USA) for Illumina^®^ in accordance with the manufacturer’s recommendations, and added index codes to attribute sequences to each sample. Then, we clustered the index-coded samples on a cBot Cluster Generation System via TruSeq PE Cluster Kit v3-cBot-HS (Illumina) in line with the manufacturer’s instructions. After cluster formation, the library preparations were sequenced on an Illumina HiSeq platform, and generated 150 bp paired-end reads. The index of the reference genome (Database: Eensemble_90, version: Gallus_gallus.Gallus_gallus-5.0.dna.toplevel.fa) was established using Bowtie v2.2.3, and the high-quality RNA-seq reads were aligned to the reference genome using the HISAT2 v2.0.4 [23] program with default parameters. Then, we used HTSeq v0.6.1 to compute the read numbers mapped to each gene. Following this, we calculated the fragments per kilobase of transcript per million mapped reads (FPKM) value of each gene according to the length of the gene and the read count numbers (using Perl script). The FPKM value of the genes was conducted to analyze the transcriptional level of the genes in the A and B compartments.

### 2.8. Enrichment Analysis

For genes annotated in switched compartments A/B, we used KOBAS 2.0 [24] to examine statistically significant enrichment in KEGG pathways. After that, gene ontology (GO) enrichment analysis was implemented via the GOseq R package [25]. Here, the genes identified in 5% switched A/B compartments between LDC and WCC liver cells were defined as switched compartmental genes (SCGs), and pathways with a *p*-value less than 0.05 were considered significant.

## 3. Results

### 3.1. An Integrated Map of Chromosomal Interfaces in Chicken Liver Cell Nuclei

To analyze the dynamic chromatin interactions in the primary chicken liver cells from WCC and LDC, we conducted Hi-C experiments and sequenced approximately 295 Gb of high-quality raw data. The alignment of the procured sequences to the chicken reference genome (ftp://ftp.ensembl.org/pub/release-90/fasta/gallus_gallus/dna/, accessed on 12 May 2020) resulted in an average of 263 million paired-end reads mapped for each individual, among which an average of 207 million were intra-chromosomal reads (207,914,357 in LDC and 206,471,795 in WCC) (Appendix A). The data processing also revealed the high quality of the Hi-C data, which confirmed the successful performance of the Hi-C experiments.

### 3.2. Identification and Characterization of Compartments in Chicken Liver Cells

After data analysis, we acquired chromatin interaction heatmaps at 40 kb resolution for each sample (Figure 1 and Appendix A). The heatmaps showed a typical plaid pattern, which was previously observed in data obtained for chicken fibroblasts and erythrocytes in Hi-C [13] as well as in mammalian Hi-C data [12]. The plaid pattern intimates the existence of large spatial compartments, including compartments A/B. We are aware that the genome is composed of actively transcribed compartments A and inactive compartments B [11], and the switching of compartments A/B is related to comparative changes in gene expression [14]. Next, the genome compartment types were determined at 1 Mb resolution within liver cells and it was found that masses of genomic regions reserved the same compartments within the LDC liver cells compared with those in the WCC liver cells (Figure 2A,B). A total of 5% genomic regions switched between the compartments A and B, and as anticipated, more genes were detected in compartment A, which showed higher transcriptional levels than the genes in compartment B (Figure 2C–F and Appendix A).

We next compared the TADs in the LDC and WCC liver cells. The TADs from Hi-C interaction matrices were called at 40 kb resolution, and 396 and 386 TADs were recognized in LDC and WCC liver cells, respectively (Figure 3 and Appendix A), with median TAD sizes of 2.3 Mb (Appendix A). Interestingly, previous studies have revealed that TAD boundaries showed a prominent abundance of active genes in comparison with stochastically sampled genomic regions, which suggested a potential connection between TAD formation and gene transcription [26]. We thus investigated the TAD boundary distribution and identified 280 and 261 TAD boundaries in the LDC and WCC genomes, respectively (Appendix A), with enriched TSS in the TAD boundary (Appendix A). These findings further confirmed the reliability of the sequencing results.

### 3.3. Enrichment Analysis of SCGs in Switched A/B Compartments

We then explored the functional influences of 3D genome changes between the LDC and WCC liver cells. GO analysis was accomplished with genes lying in the 5% genomic regions that switched between compartments B and A in either direction (A ≥ B or B ≥ A) in both the LDC and WCC liver cells. Those genes were found to cluster in two primary categories including biological process and molecular function (Figure 4A). The pathway enrichment analysis within the 5% genomic regions revealed that these genes were forcefully in connection with extreme environment adaptation-related pathways, including tight junction [27,28], notch signaling [29,30,31], vascular smooth muscle contraction [32], and RIG-I-like receptor signaling [33,34] (Figure 4B and Table 1). Within the consistent A/B compartment-switching regions, a series of genes related to cell proliferation and differentiation were identified in the tight junction, notch signaling, and RIG-I-like receptor signaling pathways. Likewise, it is worth noting that several serine–threonine kinases, such as myosin light chain kinase (*MYLK*) and rho-associated coiled-coil kinase isoform 2 (*ROCK2*) in the vascular smooth muscle contraction pathway, were found to be enriched in the vascular smooth muscle contraction pathway.

## 4. Discussion

The application of chromosome conformation capture-based high-throughput methods has significantly aided the presentation of spatial chromatin interaction architecture within mammals [35,36]. Findings from previous studies on poultry have shown the higher-order chromatin structure via Hi-C [13]. In this study, we performed a high-resolution genome-wide analysis of chromatin interactions within chicken liver cells. Our results indicate that the spatial genome organization observed in chicken cells has a typical plaid pattern without a sharp transition between the A and B compartments, which appeared to be similar in several previously investigated mammalian species [12,18,22]. In chicken, we provided effective evidence of genome partitioning in the A/B compartments in both LDC and WCC, which was in accordance with the findings in chicken erythrocytes and fibroblasts [13].

At the TAD scale, LDC hepatocyte genomes were found to involve a similar number of TADs and average TAD size compared with the WCC hepatocytes. This finding was inconsistent with findings from an earlier study between chicken erythrocytes and fibroblasts [13]. The difference could be attributed to the markedly different cellular properties. Consistent with this, the heterogeneity of cancer cells could play to more diverse 3D genomes and increase the number of TADs detected in contrast to that in normal cells [37]. We also characterized the TAD boundaries. Most of the same TAD boundaries between the two cells were identified, which further confirmed the accumulation of minor, but not sharp, genomic genetic variations that contributed to extreme environmental adaptation [8,10,38]. Additional research on 3D chicken genomes will help to further elucidate the relationship between 3D genome organization and genome alterations.

In cells undergoing specific biological phenomena, such as cell differentiation, cancer development, or response to stimuli, the 3D architecture of the genome is recombinational and associated with variations in gene expression and epigenetics [14,39,40,41]. Approximately 5% of genome regions showed switching between the A and B compartments in the comparative analysis between the LDC and WCC 3D genomes, which was linked to the influences in gene expression. Likewise, pathway enrichment analysis illustrated that the expression of some genes within the switched compartments was forcefully connected with cell proliferation and differentiation, a common adaptive response to extreme environments [42,43]. Not surprisingly, the tight junction and RIG-I-like receptor signaling pathways had been confirmed to exhibit a close connection with heat tolerance, with a fusion between immune and metabolic responses to ensure energy balance and permit growth and defense [28,33]. Interestingly, the notch signaling pathway was found to be involved in not only a high-temperature response [33], but also cold challenge [30,31], and played an extremely important role in the adaptation to hypoxic environments [30]. Vascular smooth muscle contraction, which includes vasoconstriction and vasodilation, contributes to cold acclimatization [32]. Here, we presented a list of cold adaptation candidate genes in vascular smooth muscle contraction, among which *ROCK2* and *MYLK* were found to exacerbate vasoconstriction by straightly phosphorylating myosin light chains [44,45]. Under these conditions, the validated interactions between *ROCK2* and *MYLK* further gave evidence of their role in environment adaptation [46]. These genes in our selected pathways could be major targets for tropical and frigid environment tolerance, and several of the responses can be explained with further function analysis. Nonetheless, the mechanisms facilitating adaptations to extreme environmental conditions are expected to be complex, and our findings partly provide insight into extreme environmental adaptations.

## 5. Conclusions

In summary, the findings from this study provide evidence that extreme environmental conditions altered the A/B compartments and reorganized the TADs and TAD boundaries in chicken 3D genome architectures. Moreover, tight junction, the notch signaling pathway, vascular smooth muscle contraction, and the RIG-I-like receptor signaling pathway were suggested to contribute to extreme environmental adaptation.

## Figures and Tables

**Figure 1 genes-13-02317-f001:**
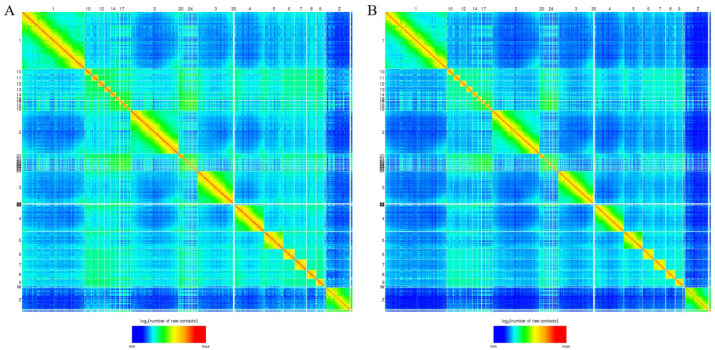
Hi-C contact heatmaps of liver cells in LDC (**A**) and WCC (**B**). The color of each dot on the heatmaps represents the log of the interaction probability for the corresponding pair of genomic loci according to standard JuiceBox color scheme.

**Figure 2 genes-13-02317-f002:**
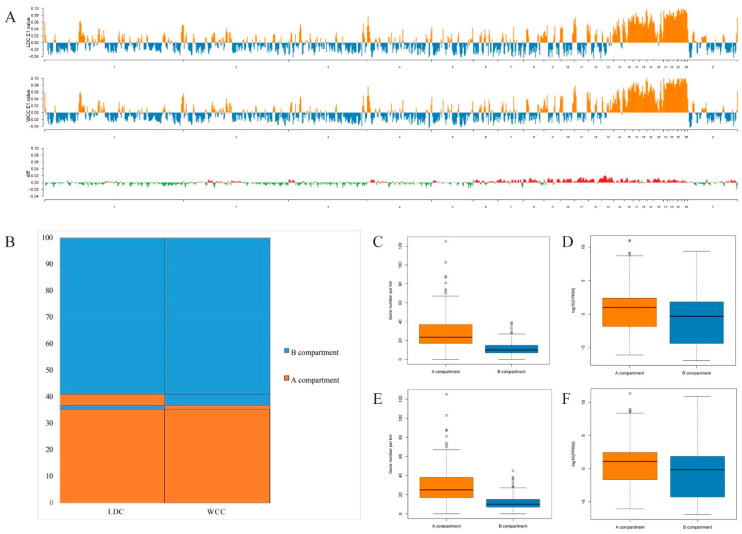
Association between A/B compartment switches and gene expression. (**A**) Distribution of the A/B compartments in the whole genome of two liver cells. A compartments are shown in orange, B compartments are shown in blue. The diff means the absolute difference between the PC1 value of two Hi-C data, and the value greater than 0 are shown in red, less than 0 are shown in green. (**B**) Genome-wide proportions of A/B compartment changes in the whole genome of two liver cells. Gene numbers (**C**,**E**) and expression (**D**,**F**) volume map of A/B compartments in the whole genome of LDC and WCC liver cells.

**Figure 3 genes-13-02317-f003:**
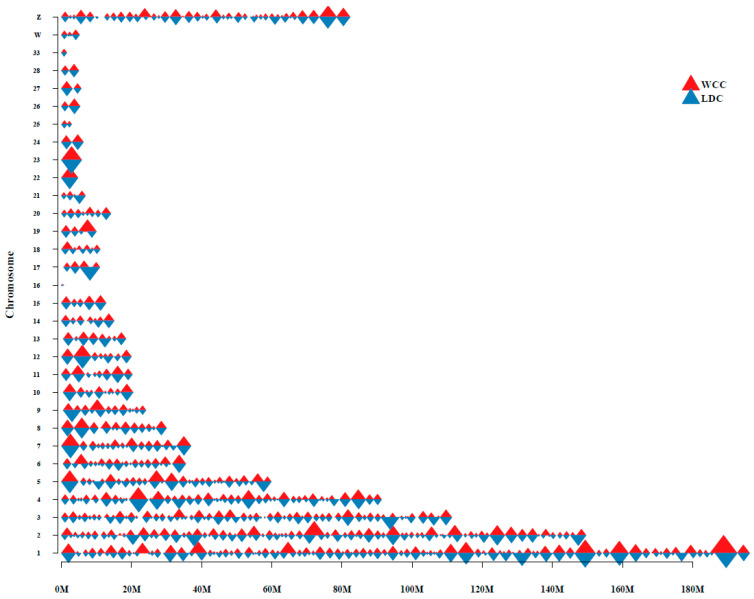
Distribution of topologically associated domains (TADs) on chromosome.

**Figure 4 genes-13-02317-f004:**
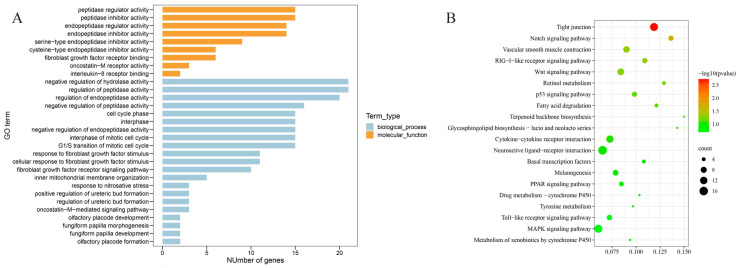
Enrichment analysis of switched compartmental genes (SCGs) in 5% switched A/B compartments between LDC and WCC liver cells. (**A**) GO enrichment analysis. The 30 most common GO terms are presented. (**B**) KEGG enrichment analysis. The 20 most common KEGG pathways are presented. The *y*-axis and *x*-axis indicate pathway name and rich factor, respectively. The size of the circle dot means gene number.

**Table 1 genes-13-02317-t001:** Enriched KEGG pathways of switched compartmental genes (SCGs) in 5% switched A/B compartments between LDC and WCC liver cells.

Term	Count	Enriched Genes *	*p*-Value
Tight junction	9	*EXOC3, PRKCQ, AMOTL1, CTTN, TJAP1, CLDN16, CLDN1, PRKCH, GNAI2*	0.00066
Notch signaling pathway	4	*MAML2, ADAM17, JAG2, DTX3L*	0.00192
Vascular smooth muscle contraction	8	*ADCY5, MYLK, KCNMB1, PRKCQ,* *PRKCH, ROCK2, MYLK4, GUCY1A2*	0.02457
RIG-I-like receptor signaling pathway	4	*IFNK, IL8, MAP3K1, FADD*	0.03433

* *EXOC3*, exocyst complex component 3; *PRKCQ*, protein kinase C theta; *AMOTL1*, angiomotin-like protein 1; *CTTN*, cortactin; *TJAP1*, tight junction-associated protein 1; *CLDN16*, claudin-16; *CLDN1*, claudin-1; *PRKCH*, protein kinase C η; *GNAI2*, G-protein α inhibiting activity polypeptide 2; *MAML2*, mammalian mastermind-like 2; *ADAM17*, a disintegrin and metalloprotesase 17; *JAG2*, jagged-2; *DTX3L*, deltex-3-like; *ADCY5*, adenylate cyclase 5; *MYLK*, myosin light chain kinase; *KCNMB1*, calcium-activated potassium channel subunit *β*-1; *ROCK2*, rho-associated coiled-coil kinase isoform 2; *MYLK4*, myosin light chain kinase 4; *GUCY1A2*, guanylate cyclase soluble subunit *α*-2; *IFNK*, interferon kappa; IL8, interleukin8; MAP3K1, mitogen-activated protein kinase kinase kinase 1; *FADD*, Fas-associated protein with death domain.

## Data Availability

The Hi-C data and transcriptome data presented in this study are in NCBI SRA (PRJNA817871) and SRA (PRJNA800119), respectively.

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
