# Peer review of "Three-Dimensional Organization of Chicken Genome Provides Insights into Genetic Adaptation to Extreme Environments"

_genes, 2022, doi:10.3390/genes13122317_

Round 1

Reviewer 1 Report

The manuscript entitled “3D organization of chicken genome provides insight into genetic adaption to extreme environments”. The tropic of this study is pretty well and the analysed methods were also standard. However, for further improvement, I have the following comments.

1. Why select the liver RNA to be sequenced. Is the liver important for climate adaptation?

2. L61-62, why selected the chickens at different ages, 42 weeks LDCs and 43 weeks WCCs?

3. Try to avoid using I and We in the scientific writing.

4. Unify the formats (eg.: number in digits or letter format).

5. Some references are not completed as bibliography.

6. L296 In conclusion, the author pointed out that glycosphingolipid biosynthesis pathway was important, however, this pathway didn’t mention before, the same as in the abstract.

7. The English language of this paper need polished. Some mistakes were appeared in this paper.

L12, remain should be remains; L22, their should be its; L73, were should be was; L108, dividing should be divided; L121, among should be between; L137, should be ‘a significant correlation’; L211, were should be was, lain should be lying; L246, significant should be significantly; L264, farther should be further; LL276 an should be a; L285, adaption should be adaptation.

Reviewer 2 Report

This work adapted high-throughput chromosome conformation capture (Hi-C) to interrogate the three-dimensional(3D) genome architecture differences of liver tissues of two chicken breeds, and the potential functional implications of 3D genome architecture on the adaption to extreme environments. To this end, the authors generated Hi-C and RNA-seq data for integrative analysis.  This work is interesting and well-motivated, and the data can serve as a good resource for genomic and epigenomic studies. However, several issues stand out which are listed as follows.

From lines 60-63, it states both Wenchang and Lindian chickens were from the “national natural poultry conservation farm with tropical and frigid environmental zones”, in the meantime, it states that one Wenchang chicken was from Wenchang, Hainan; and the Lindian chicken was from Lindian, Heilongjiang. Presumably,  this means two experimental subjects are of different breeds and may experience distinct living environments. This is not detrimental to the whole study, but detailed housing history and conditions should be provided for the cautious interpretation of the 3D-genome and transcriptomic changes discovered thereafter.

In the Methods, the references of multiple computational tools used in this study have not been properly provided, such as hicPipe, hiclib, HISAT2, KOBAS, and GOseq. A major part of this work is on the switch between A/B compartments, hence more information about this analysis should be provided. Figure S2 shows that the authors use gene numbers to set the active A compartments from inactive B compartments, which should be added in the Method description. Also, the lower panel of Figure 2 which might show the absolute difference between the PC1 value of two Hi-C data, is not been described in methods, figure legend or main text. This could be misleading because the functional analysis in section 3.3 was focused on A à B, B à switches but not this type of “diff” in Figure 2. In addition, I suggest that Figures 2, 3, and 4 could be integrated into one figure for conciseness given that they are all descriptive results about the A/B compartments.

In lines 198-207, the authors identified TADs in LDC and WCC, however, the results of TADs boundary correlation and “cell type-specific” boundary mentioned in line 136 are not provided. Further, the possible transcriptional implication of TADs boundary should be analyzed with RNA-seq data.

The GO terms in figure 6 are not properly displayed. And similarly, figure 6 and figure 7 can be integrated as one figure.

In section 3.3, The authors argue that chromatin conformational change is associated with gene expression change.  However, the description of this section is confusing: all functional annotations (GO analysis and KEGG analysis) are performed on “genes lain in 5% genomic regions which switched between compartments B and A in either direction…” (lines 212-214), while the title of Figure 6 and Table 1 are “GO enrichment analysis of DEGs in different A/B compartments between LDC and WCC liver cells.” and “Enriched KEGG pathways of DEGs of different A/B compartments between LDC and WCC  liver cells.” The authors should explain the definition of “DEGs”  which has not been introduced before.

Likewise, the “G4” and “AG4-1” is not properly annotated in Table S4 which makes readers hardly comprehend.

The wording and nomenclature of this article should be improved. For example, “High-through” in line 10 should be “High-throughput”; and “divided” in line 82 should be “captured”, “collected”, or “enriched”; “constructing” in lines 220 and 246 should be “conformation”; etc…

Round 2

Reviewer 2 Report

Thanks to the authors' efforts, this manuscript has been largely improved. However, it is not satisfactory for one issue which needs to be further addressed.

Regarding the definition of “DEG”, which is provided by the author in Line 173: “Here, genes and pathways with P-value less than 0.05 were recognized as significant, and the significantly different expression genes in 5% switched A/B compartments between LDC and WCC liver cells were defined as DGEs”; line 256: “Enrichment analysis of DEGs in 5% switched A/B compartments between LDC and WCC liver cells”  and in the rebuttal letter: “Hence the definition of “DEGs” here represented the significantly different expression genes in 5% switched A/B compartments between LDC and WCC liver cells” are still confusing to me. Could you please help clarify?

Take the sentence in line 173 as an example: here “DGE” is an obvious typo for DEG which is usually used as the abbreviation of the differentially expressed gene. “the significantly different expression genes in 5% switched A/B compartments between LDC and WCC liver cells were defined as DGEs”: To me, this sentence suggests that a series of analyses have been done, firstly “different expression genes” were identified by differential analysis of RNA-seq data, secondly; “5% switched A/B compartments were identified; finally, a portion of the “different expression genes” located in the 5% switched A/B compartments  “were defined as DGEs”. It seems an intersection between “different expression genes” and genes in “5% switched A/B compartments” has been performed, and the shared genes “were defined as DGEs”. However,  all above is only my speculation and there is no information provided by the authors about the differential analysis of RNA-seq data, lists of DEGs identified by RNAseq, nor the genome coordinates of flipped A/B compartments.

Or, the DEG is not the abbreviation of the differentially expressed gene used by most researchers, but rather, it refers to genes in “5% switched A/B compartments”.

In summary, if DEGs were identified by the intersection between RNAseq DEGs and genes in 5% switched A/B compartments, the authors should consider providing the whole lists of RNA-seq DEGs, 5% A/B compartments regions, and the intersected final DEGs located in 5% A/B compartments. If the identification of DEGs is irrelevant to RNA-seq, then another abbreviation can be considered for these genes, such as SCG for “switched compartmental genes” for clarification. 
